# Sodium Houttuyfonate Prevents Seizures and Neuronal Cell Loss by Maintaining Glutamatergic System Stability in Male Rats with Kainic Acid-Induced Seizures

**DOI:** 10.3390/biomedicines12061312

**Published:** 2024-06-13

**Authors:** Yi Chang, Yi-Jun Chen, Su-Jane Wang

**Affiliations:** 1School of Medicine, Fu Jen Catholic University, New Taipei City 24205, Taiwan; m004003@ms.skh.org.tw; 2Department of Anesthesiology, Shin Kong Wu Ho-Su Memorial Hospital, Taipei 11101, Taiwan; 3Department of Respiratory Therapy, Fu Jen Catholic University, New Taipei City 24205, Taiwan; serein2237@gmail.com; 4Research Center for Chinese Herbal Medicine, College of Human Ecology, Chang Gung University of Science and Technology, Taoyuan City 33303, Taiwan

**Keywords:** sodium houttuyfonate, antiseizure, glutamate, kainic acid, hippocampus

## Abstract

The present study evaluated the antiseizure and neuroprotective effects of sodium houttuyfonate (SH), a derivative of *Houttuynia cordata* Thunb. (*H. cordata*), in a kainic acid (KA)- induced seizure rat model and its underlying mechanism. Sprague Dawley rats were administered normal saline, SH (50 or 100 mg/kg), or carbamazepine (300 mg/kg) by oral gavage for seven consecutive days before the intraperitoneal administration of KA (15 mg/kg). SH showed antiseizure effects at a dose of 100 mg/kg; it prolonged seizure latency and decreased seizure scores. SH also significantly decreased neuronal loss in the hippocampi of KA-treated rats, which was associated with the prevention of glutamate level increase, the upregulation of glutamate reuptake-associated proteins (excitatory amino acid transporters 1–3), glutamate metabolism enzyme glutamine synthetase, the downregulation of the glutamate synthesis enzyme glutaminase, and significant alterations in the expression of AMPA (α-amino-3-hydroxy-5-methyl-4-isoxazole-propionic acid receptor) and NMDA (N-methyl-D-aspartic acid receptor) receptor subunits in the hippocampus. Furthermore, the effects of SH were similar to those of the antiseizure drug carbamazepine. Therefore, the results of the present study suggest that SH has antiseizure effects on KA-induced seizures, possibly through the prevention of glutamatergic alterations. Our findings suggest that SH is a potential alternative treatment that may prevent seizures by preserving the normal glutamatergic system.

## 1. Introduction

Epilepsy, a common neurological disease that affects approximately 70 million people globally, is characterized by spontaneous and recurrent seizures due to the excessive discharge of cerebral neurons [1]. There are currently more than 25 antiseizure drugs that act by increasing GABAergic inhibitory neurotransmission, reducing glutamatergic excitatory neurotransmission, or blocking ion channels [2]. However, the long-term use of current antiseizure drugs has limited efficacy and undesirable side effects [3]. Accordingly, the discovery of a new antiseizure drug with increased efficacy and an improved safety profile is of fundamental importance. In this context, some plant extracts and products can potentially alleviate convulsions or seizures; therefore, natural products are important sources of new antiseizure drugs [4].

*Houttuynia cordata* Thunb. (*H. cordata*) is commonly used in traditional Chinese medicine as an antipyretic and detoxifying herbal medicine for the treatment of inflammatory-related diseases and infections [5,6]. Sodium houttuyfonate (SH, C12H23O5SNa, MW 302.36; Figure 1A) is a derivative of *H. cordata* and has various pharmacological properties, including anti-inflammatory, antibacterial, antiviral, and cardiovascular protective effects [7,8,9]. In addition, there is evidence that SH can ameliorate neuronal damage, neuroinflammation, and memory impairment in Aβ1-42-induced Alzheimer’s disease (AD) model mice [10]. Neuroprotective effects were also demonstrated in traumatic brain injury in mice [11] and in spinal cord injury in rats [12]. However, regarding the anticonvulsant property of SH, no studies have directly linked this component to this effect.

An excessive inflammatory response in the brain has been shown to hyperexcite neurons and lead to neuronal damage, which results in permanent impairment of the structure and function of neural networks, an important underlying cause of the recurrent spontaneous seizures observed in epilepsy [13,14]. Additionally, anti-inflammatory agents can alleviate epileptogenesis and seizures [15,16,17]. Given the promising anti-inflammatory property of SH, its potential therapeutic role in epilepsy should be considered. Therefore, the aim of this study is to evaluate the effect of SH on seizures in a kainic acid (KA)-induced seizure model in rats and compare its effects with those of antiseizure drug carbamazepine (CBZ).

## 2. Materials and Methods

### 2.1. Drugs and Chemicals

SH (98% purity) was obtained from ChemNorm (Wuhan, China). KA, CBZ, and all other reagents were purchased from Sigma-Aldrich (St. Louis, MO, USA).

### 2.2. Animals

Male Sprague Dawley rats (170–200 g; BioLASCO, Taipei, Taiwan) were housed in standard cages. The animal room was under controlled conditions of temperature (22 ± 2 °C) and was under a 12/12 h light/dark cycle. The rats had free access to food, chow, and water. The rats were left undisturbed for a minimum of 3 days to allow adaptation to the new environment. The rats were weighed daily in the morning, after which body weights were recorded. All procedures were performed in accordance with the National Institutes of Health Guide for the Care and Use of Laboratory Animals (NIH Publications No. 80-23) and were approved by the Committee for Animal Use of the Fu Jen Catholic University (project number A11057).

### 2.3. Seizure Model and Drug Treatment

Seizures were induced by intraperitoneal (i.p.) administration of 15 mg/kg KA to rats [18,19,20,21]. Rats were randomly subdivided into 4 groups (each containing 9–11 rats) as follows: (a) control group: rats received a 0.3 mL normal saline containing 5% polyethylene glycol oral gavage for 7 days (once per day); (b) KA group: rats received a 0.3 mL normal saline containing 5% polyethylene glycol oral gavage for 7 days and a KA injection (15 mg/kg in 0.3 mL saline, i.p.) on the 8th day; (c) SH + KA group: rats received an SH oral gavage (50 or 100 mg/kg in 0.3 mL saline containing 5% polyethylene glycol) for 7 days and a KA injection (i.p.) on the 8th day; and (d) CBZ + KA group: rats received a CBZ oral gavage (100 mg/kg in 0.3 mL saline) for 7 days and a KA injection (i.p.) on the 8th day. After each KA injection, each rat was placed in a cage to record its seizure behavior for 3 h. The latency to tonic–clonic seizure onset (min) and seizure score were recorded based on Racine’s scale as follows: state 0: no response; state 1: ear and facial twitching; state 2: myoclonic jerks; state 3: myoclonic jerks, rearing; state 4: turning over onto the side position, tonic–clonic seizures; state 5: turning over onto the back position, generalized tonic–clonic seizures [22]. On the 11th day, the rats were sacrificed for subsequent experiments (Figure 1B). The doses and schedules for drug administration were selected based on previous studies and pilot experiments [19,21].

### 2.4. Nissl Staining and NeuN Immunofluorescence

The rats were euthanized by administering Zoletil (40 mg/kg i.p.; Virbac, Carros, France) under deep anesthesia. Subsequently, the rats were transcardially perfused with 100 mL of saline (0.9%). Thereafter, the rats were perfused with 50 mL of paraformaldehyde prepared in 0.05 M sodium phosphate (pH 7.4, containing 0.8% saline). The brains were isolated, fixed in 4% paraformaldehyde at 4 °C for 24 h, and transferred to 30% sucrose solution for 7 days at 4 °C. The frozen brains were postfixed in the same perfusion fixative and then washed and embedded in paraffin to prepare paraffin blocks, which were sectioned into 30 µm thickness coronal slices using a frozen microtome.

Nissl staining was performed to detect neuronal damage; this is a classic nucleic acid staining method for nervous system tissue. Briefly, the brain slices were mounted onto gelatin-coated slides, rehydrated in distilled water, and then transferred to a solution of 0.5% of cresyl violet in 0.1% acetic acid (Abcam, Cambridge, UK) for 10 min. The slides were washed in distilled water, dehydrated in ethanol-graded solutions (70%, 95%, and 100%), cleared in xylene, and finally covered with dibutyl phthalate in xylene medium (DPX, Sigma-Aldrich).

NeuN immunofluorescence was conducted to examine intact neurons. In brief, brain sections were rinsed in distilled water and treated with 1% hydrogen peroxide for 15 min to remove endogenous peroxidase activity. Then, the brain sections were incubated overnight at 4 °C with a mouse anti-NeuN antibody (1:1000), washed in distilled water, and incubated with the anti-mouse secondary antibody conjugated with tetramethylrhodamine isothiocyanate (TRITC) for 2 h at room temperature. The nuclei were counterstained with DAPI (50 ng/mL). Then, brain sections were washed three times with distilled water. Finally, the free-floating brain sections were mounted on gelatin-coated glass slides, dehydrated, cleared with xylene, and covered with DPX.

Images from Nissl and NeuN were acquired using a fluorescence microscope (Zeiss Axioskop 40, Göttingen, Lower Saxony, Germany). Leica 4× or 10× objective lenses with numerical apertures (NA) of 0.1 or 0.25 were used in this study. The number of surviving neurons and NeuN-positive cells per mm^2^ was calculated in 4 consecutive coronal sections for each animal by an examiner blinded to the experimental conditions. The data were averaged for each animal using ImageJ image analysis software (version 6.0) (NIH Image, National Institutes of Health, Bethesda, MD, USA).

### 2.5. Glutamate Level

The glutamate levels in the hippocampus tissues were measured using high-performance liquid chromatography (HPLC) as described previously [20]. Briefly, rats from each group were sacrificed by decapitation, and the hippocampus was homogenized in HEPES buffer medium and centrifuged at 15,000× *g* for 10 min at 4 °C. The supernatant (10 μL) was filtered through a 0.22 µm membrane filter and injected into an HPLC instrument (HTEC-500, Eicom, Kyoto, Japan). The glutamate concentration was determined using peak areas with an external standard method and is expressed herein as ng/mg protein.

### 2.6. Western Blotting

Western blotting was performed as previously described [19]. Briefly, equal amounts of protein per sample (30 μg) were loaded on a 10% sodium dodecyl sulfate (SDS)–polyacrylamide gel electrophoresis. Samples were then transferred to nitrocellulose membranes, and 5% nonfat milk in TBST for 1 h was used for membrane blocking. After overnight incubation at 4 °C, primary antibody solutions in 1% nonfat milk in TBST were added. The primary antibodies used in this study included EAAT1 (1:10,000; Abcam, Cambridge, UK), EAAT2 (1:50,000; Abcam, Cambridge, UK), EAAT3 (1:10,000; Cell Signaling, Beverly, MA, USA), GS (1:50,000; Abcam, Cambridge, UK), glutaminase (1:10,000), GluA1 (1:5000; Invitrogen, Waltham, MA, USA), GluA2 (1:5000; Invitrogen, Waltham, MA, USA), GluN2A (1:2000; Cell Signaling, Beverly, MA, USA), GluN2B (1:2000; Cell Signaling, Beverly, MA, USA), and β-actin (1:10,000; Cell Signaling, Beverly, MA, USA). After washing with TBST, secondary antibody solutions (anti-mouse HRP-linked and anti-rabbit HRP-linked; 1:2000, Gentex, Zeeland, MI, USA) were added (1 h at room temperature). Finally, an enhanced chemiluminescence substrate (Amersham, Buckinghamshire, UK) was added and luminescence was captured. Scanning densitometry was used to quantify four to five independent experiments, which were then analyzed with ImageJ software (version 1.53) (Synoptics, Cambridge, UK). The protein/β-actin ratio was calculated.

### 2.7. Statistical Analysis

The results were plotted using GraphPad Prism version 8 (GraphPad Software, Inc., La Jolla, CA, USA) and expressed as mean ± standard error of the mean (SEM). Multiple comparisons were performed using a one-way analysis of variance (ANOVA) followed by Tukey’s post hoc test. Values of *p* < 0.05 were considered significant.

## 3. Results

### 3.1. Seizure Onset, Seizure Score, and Percentage of Animals That Developed Seizures

To explore the effect of SH on KA-induced seizures, rats were orally administered 50 or 100 mg/kg SH for 7 days before KA injection (15 mg/kg, i.p.). As shown in Table 1, the seizure latency and seizure score were significantly different in the group administered 100 mg/kg SH compared to the group administered KA alone (*p* < 0.001). Moreover, the effects of 100 mg/kg SH on seizure onset time and seizure score were similar to those of 100 mg/kg CBZ (*p* > 0.05). Compared to the KA group, 50 mg/kg SH significantly prolonged seizure latency (*p* < 0.001) but did not significantly decrease the seizure score (*p* = 0.001). In addition, the percentage of animals that developed seizures was dramatically lower in the SH pretreatment group than in the KA group. These results suggested that SH could prevent KA-induced seizures. Since 100 mg/kg SH was more effective, we applied 100 mg/kg SH in further assays unless otherwise described.

### 3.2. Histological Examination

We subsequently assessed whether the antiseizure effect of SH on KA-induced seizures was associated with a protective effect against excitotoxic cell death. Figure 2A shows the results obtained from Nissl staining of brain sections containing the anterior hippocampus. As shown in Figure 2A–C, the control group exhibited normal neuronal shape and number in the CA1 and CA3 regions of the hippocampus. Three days after the KA injection, the regular arrangement of neurons was disrupted, and the number of neurons in the CA1 and CA3 regions of the hippocampus was reduced in the KA group compared to the control group (*p* < 0.0001). Conversely, preadministration of 100 mg/kg SH or 100 mg/kg CBZ significantly attenuated these alterations in the CA1 and CA3 regions, respectively (*p* < 0.0001). A similar protective effect of SH was observed through labeling with the neuron marker NeuN. As shown in Figure 3A–C, a marked decrease in the number of NeuN-positive neurons in the CA1 and CA3 regions was observed in the KA group compared to the control group (*p* < 0.0001). However, the number of NeuN-positive neurons was significantly greater in the SH and CBZ + KA groups than in the KA group (*p* < 0.0001). No significant difference was observed between the SH group and the CBZ group (*p* > 0.05). The number of neurons and NeuN-positive neurons in the CA2 region was also reduced in the KA group compared to the control group (*p* < 0.0001; Appendix A). Neuronal survival and NeuN-positive neurons in the CA2 of SH + KA-treated rats were also significantly higher than that observed in KA-treated rats (*p* < 0.0001; Appendix A). In addition, the numbers of living and NeuN-positive neurons in the DG region did not differ in all groups (*p* > 0.05; Appendix A). The SH-only group showed no obvious neuronal loss in the CA1 and CA3 regions compared to the control group (*p* = 1) but was significantly different from the KA group (*p* < 0.0001) (Figure 2 and Figure 3). These results suggested that SH could prevent hippocampal cell death in KA-injected rats, corresponding to the ability of SH to alleviate seizures.

### 3.3. Glutamate Level

Excess glutamate in the brain contributes to the pathogenesis of epilepsy [23]. To investigate whether SH might modulate the glutamatergic system, contributing to its antiseizure and neuroprotective effects, we measured the concentration of glutamate in the hippocampus of the rats by HPLC. Glutamate levels were markedly greater in the hippocampus of the KA group than in that of the control group (*p* < 0.0001). However, pretreatment of KA-treated rats with SH (100 mg/kg) or CBZ (100 mg/kg) resulted in a significantly lower glutamate concentration than that in the KA group (*p* < 0.0001), and the glutamate concentration was not different from that in the control group (*p* = 1) (Figure 4). No significant difference was observed in the glutamate concentration between the SH- and CBZ-treated groups (*p* = 0.99). These results suggest that SH can attenuate KA-induced seizure and hippocampal cell death by inhibiting excessive glutamate signals in the hippocampus.

### 3.4. EAAT1, EAAT2, EAAT3, GS, and Glutaminase Levels

We then evaluated whether the SH-induced decrease in glutamate levels was associated with the altered expression of glutamate reuptake-related proteins (EAAT1, EAAT2, and EAAT3) and glutamate metabolism-related enzymes (GS and glutaminase) in the hippocampus by Western blotting. As shown in Figure 5A–F, compared to those in the control group, the protein expression levels of EAAT1, EAAT2, EAAT3, and GS were significantly decreased (*p* < 0.0001). In contrast, glutaminase protein expression was significantly increased in the hippocampus of the KA group (*p* < 0.0001). In the rats that received SH (100 mg/kg) or CBZ (100 mg/kg) before KA injection, the EAAT1, EAAT2, EAAT3, and GS protein levels increased (*p* < 0.0001), while the glutaminase protein level decreased compared to those in the KA group (*p* < 0.0001). No significant difference was observed between the SH group and the CBZ group (*p* > 0.05). These results suggest that SH can preserve the normal levels of EAAT1, EAAT2, EAAT3, GS, and glutaminase in the hippocampi of kainic acid-treated rats.

### 3.5. GluA1, GluA2, GluNR1, GluN2A, and GluN2B Levels

Since alterations in glutamate receptor subunit compositions have been observed in seizure animal models [24,25], we assessed the expression levels of the main subunits of AMPAR (GluA1 and GluA2) and NMDAR (GluN2A and GluN2B) in the hippocampus by Western blot analysis. As shown in Figure 6A–E, the protein expression levels of GluA1, GluA2, and GluN2A in the hippocampus were decreased (*p* < 0.0001), while that of GluN2B was increased in the KA group compared to the control group (*p* < 0.0001). In contrast, compared with those in the KA group, the GluA1, GluA2, and GluN2A expression levels in the SH (100 mg/kg) and CBZ (100 mg/kg) + KA groups significantly increased, and the GluN2B expression levels significantly decreased (*p* < 0.0001). No significant difference was observed between the SH group and the CBZ group (*p* > 0.05). These results suggest that the protective effect of SH on KA-induced seizure and hippocampal cell death might be associated with alterations in AMPAR and NMDAR subunit compositions in the hippocampus.

## 4. Discussion

Studies using in vivo models of neurological damage have shown that SH is an efficient and promising natural neuroprotective agent [10,12]. In the present study, we investigated the effects of administering a single high dose of SH prior to KA-induced seizures on latency time, severity, glutamate level, and hippocampal damage associated with the seizures.

KA is an ionotropic glutamate receptor subtype agonist that induces the influx of cellular calcium, mitochondrial dysfunction, neuronal apoptosis, and degeneration [26]. The systemic administration of KA in rodents is a well-known chemical-induced model of epilepsy. This model appears to be highly similar to epilepsy in humans [27]. In the present study, an i.p. injection of 15 mg/kg KA reduced the latency to develop seizures and increased the Racine seizure score in all rats, which is consistent with the results observed by other investigators [28,29]. In contrast, SH pretreatment delayed seizure onset such that the latency to develop seizures after KA injection was prolonged. In addition, decreased seizure intensity was revealed by a reduction in seizure scoring and a decreased percentage of animals that developed seizures in the SH + KA group. Furthermore, 100 mg/kg SH had a similar effect to that of the antiseizure drug CBZ. These results confirm the antiseizure activity of SH. In addition, seizures induced by KA can result in excessive neuronal loss, especially in the hippocampus [20,21,30,31]. Similarly, we found that significant hippocampal neuronal loss was observed in KA-treated rats and that this effect was reversed by SH pretreatment, suggesting that SH plays a preventive role in KA-induced seizure rats.

An important element in epilepsy is neuronal hyperactivity. Such hyperactivity was reported to be associated with increased glutamate levels. The overactivation of glutamate receptors by excess glutamate causes calcium overload, which promotes seizures [23,32]. Consistent with previous studies [20,21], we found that KA treatment increased glutamate levels in the hippocampus. SH pretreatment decreased glutamate levels in the hippocampus of KA-treated rats, thereby reducing the severity of seizures. In addition, we studied changes in the protein expression of EAAT1, EAAT2, EAAT3, GS, and glutaminase in the hippocampus of KA-treated rats and the possible role of SH. Astrocytic EAAT1, EAAT2, and neuronal EAAT3 remove extracellular glutamate, GS converts glutamate to glutamine within astrocytes, and glutaminase converts glutamine into glutamate in neurons [33,34,35]. Previous studies have shown that decreased EAAT1, EAAT2, EAAT3, and GS or increased glutaminase cause an increase in glutamate concentrations in the brain and subsequent excitotoxicity [34,36,37,38]. Similarly, we found that KA treatment significantly decreased the protein levels of EAAT1, EAAT2, EAAT3, and GS in the hippocampus, which suggests that the decreased uptake of glutamate leads to increased glutamate levels in kainic acid-treated rats. In addition, an increase in glutaminase protein levels was also observed in the hippocampi of KA-treated rats. An increase in glutaminase levels might cause the increased synthesis of glutamate, contributing to the increase in glutamate levels in KA-treated rats. These results suggest that increased synaptic glutamate synthesis and decreased glutamate clearance led to an increase in glutamate levels in the hippocampi of KA-treated rats. Furthermore, SH pretreatment decreased the protein level of glutaminase and increased the protein levels of EAAT1, EAAT2, EAAT3, and GS in the hippocampi of KA-treated rats. These findings suggest that SH preserves the normal synthesis and clearance of glutamate, a likely explanation for the decreased glutamate level in KA-treated rats. Since seizures induced by KA are due to the enhancement of the glutamatergic system, the findings in this study suggest that a decrease in glutamate levels in the brain might be a mechanism by which SH exerts its neuroprotective effects against KA-induced seizures. However, how SH regulates EAAT1, EAAT2, EAAT3, GS, and glutaminase in KA-induced seizures remains to be elucidated.

Moreover, we also observed significant alterations in the subunits of the AMPA (GluA1 and GluA2) and NMDA (GluN2A and GluN2B) glutamate receptors, including decreased GluA1, GluA2, and GluN2A levels and increased GluN2B levels, in the hippocampus after KA treatment. These alterations are consistent with the results of previous studies [20,38,39]. GluA2-containing AMPA receptors are Ca^2+^-impermeable [40]. GluN2B-containing NMDA receptors enable a greater influx of Ca^2+^ through NMDA receptors than GluN2A-containing receptors [41]. Therefore, a reduction in GluA2 levels and an increase in the expression of the GluN2B subunits can lead to an increase in the glutamate-mediated penetration of Ca^2+^ into the cell, causing excitotoxicity, cell death, and seizures [24,42,43]. Notably, in addition to preserving normal glutamate levels in the hippocampus, SH pretreatment increased the expression levels of GluA1, GluA2, and GluN2A but decreased GluN2B expression in the hippocampi of KA-treated rats. This may be related to the ability of glutamate receptors to prevent excessive Ca^2+^ influx and glutamate excitotoxicity. Therefore, we infer that SH may maintain the normal function of AMPA and NMDA receptors by regulating the expression of the GluA2, GluN2A, and GluN2B subunits, which may be involved in the neuroprotective effect of SH against KA-induced hippocampal damage and, consequently, its anticonvulsive effect. Overall, SH pretreatment increased the levels of glutamate reuptake-associated proteins (EAAT1-3) and the glutamate metabolism enzyme GS but decreased the level of the glutamate synthesis enzyme glutaminase in the hippocampi of KA-induced rats, resulting in a decrease in the glutamate concentration in the hippocampus. SH pretreatment also preserved the GluA1/GluA2-containing AMPA receptors and GluN2A-containing NMDA receptors in the hippocampus, preventing Ca^2+^ overload in neurons and KA-induced excitotoxic injury. These effects might be related to the attenuation of neuronal hyperexcitability, seizure generation, and neuronal damage in KA-treated rats (Figure 7).

CBZ is a common antiepileptic drug, and its main action is thought to be the inhibition of Na^+^ channels and synaptic glutamate release [44,45]. In the present study, the preventive effects of 100 mg/kg SH on KA-induced seizures and pathological alterations were similar to those of 100 mg/kg CBZ. This indicates that SH exerts its antiseizure effect through similar mechanisms to CBZ, particularly in the regulation of the glutamate system. In addition, although CBZ is a widely used antiepileptic drug, it has numerous side effects. SH is stable with satisfactory solubility and retains the major activities of *H. cordata* [46]. We infer that SH has the potential to be an effective and safe antiseizure drug. Future studies are essential to support this notion. On the other hand, male rats were used in our study; this is because male rats have fewer variables than female rats since hormones can modulate seizure, neuropathology, and treatment outcomes [47,48]. However, it is important to consider the influences of sex on treatment outcomes in investigational new drug studies.

## 5. Conclusions

The results of the present study suggested that SH has antiseizure effects on KA-induced seizures, possibly through the prevention of glutamatergic alterations. Our findings support the idea that SH may become a valid therapeutic intervention for the management of epilepsy.

## Figures and Tables

**Figure 1 biomedicines-12-01312-f001:**
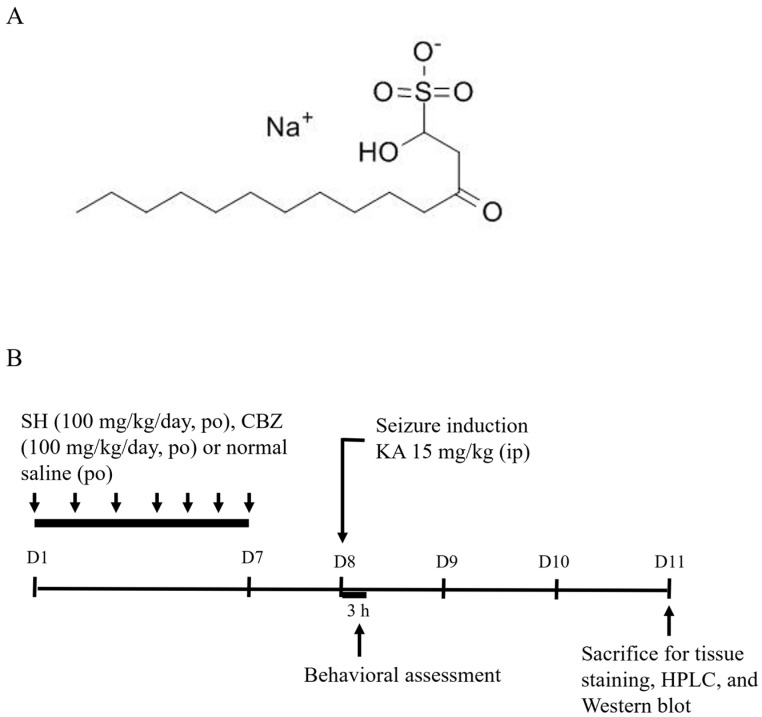
(**A**) Chemical structure of SH. (**B**) General experimental protocol.

**Figure 2 biomedicines-12-01312-f002:**
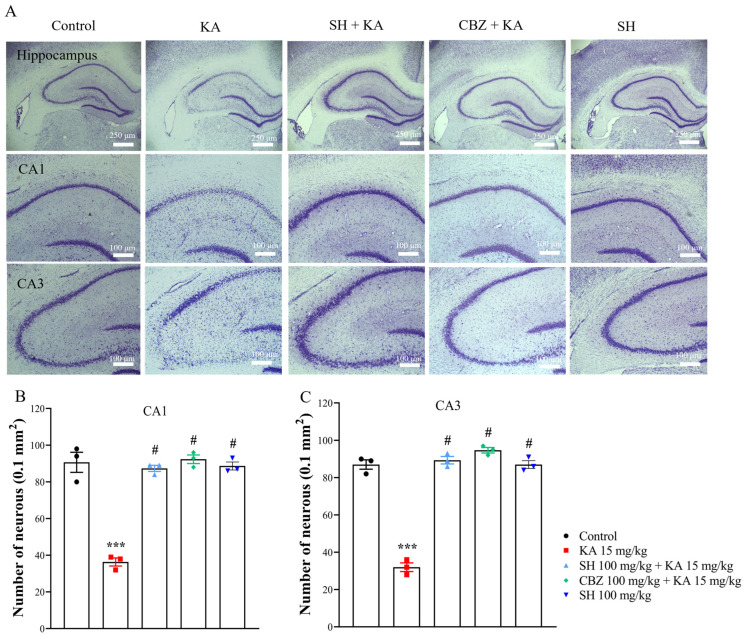
Cresyl violet staining in the hippocampal tissue of rats three days after KA injection. Representative images (**A**) and quantitative data (**B**,**C**) for hippocampal CA1 and CA3 neurons. N = 3 animals per group. *** *p* < 0.0001 compared to the control group; # *p* < 0.0001 compared to the KA group.

**Figure 3 biomedicines-12-01312-f003:**
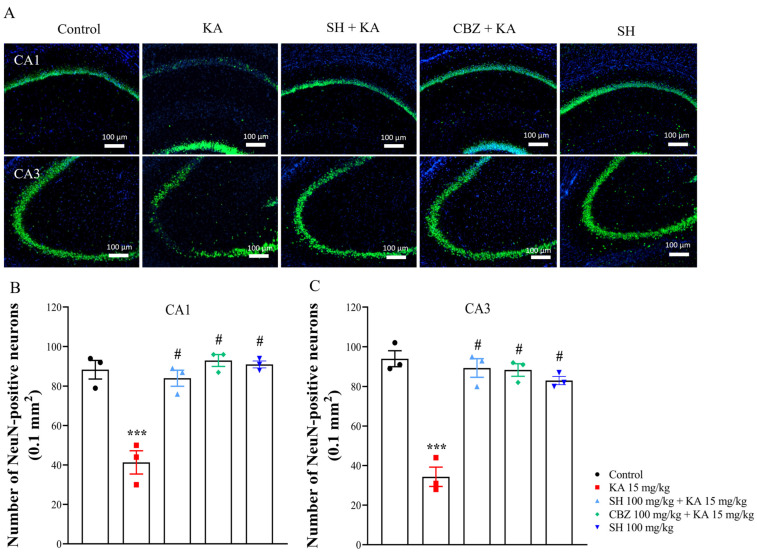
NeuN expression in the hippocampal tissue of rats three days after KA injection. (**A**) Representative images (**A**) and quantitative data (**B**,**C**) for NeuN staining. N = 3 animals per group. *** *p* < 0.0001 compared to the control group; # *p* < 0.0001 compared to the KA group.

**Figure 4 biomedicines-12-01312-f004:**
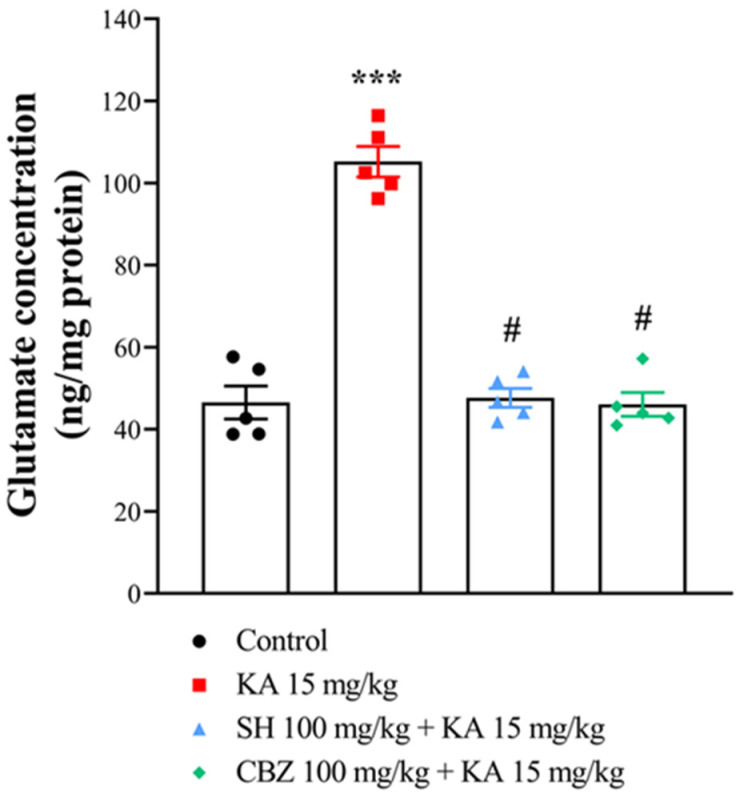
HPLC analysis of the glutamate level in the hippocampal tissue of rats three days after KA injection. N = 5 animals per group. *** *p* < 0.0001 compared to the control group; # *p* < 0.0001 compared to the KA group.

**Figure 5 biomedicines-12-01312-f005:**
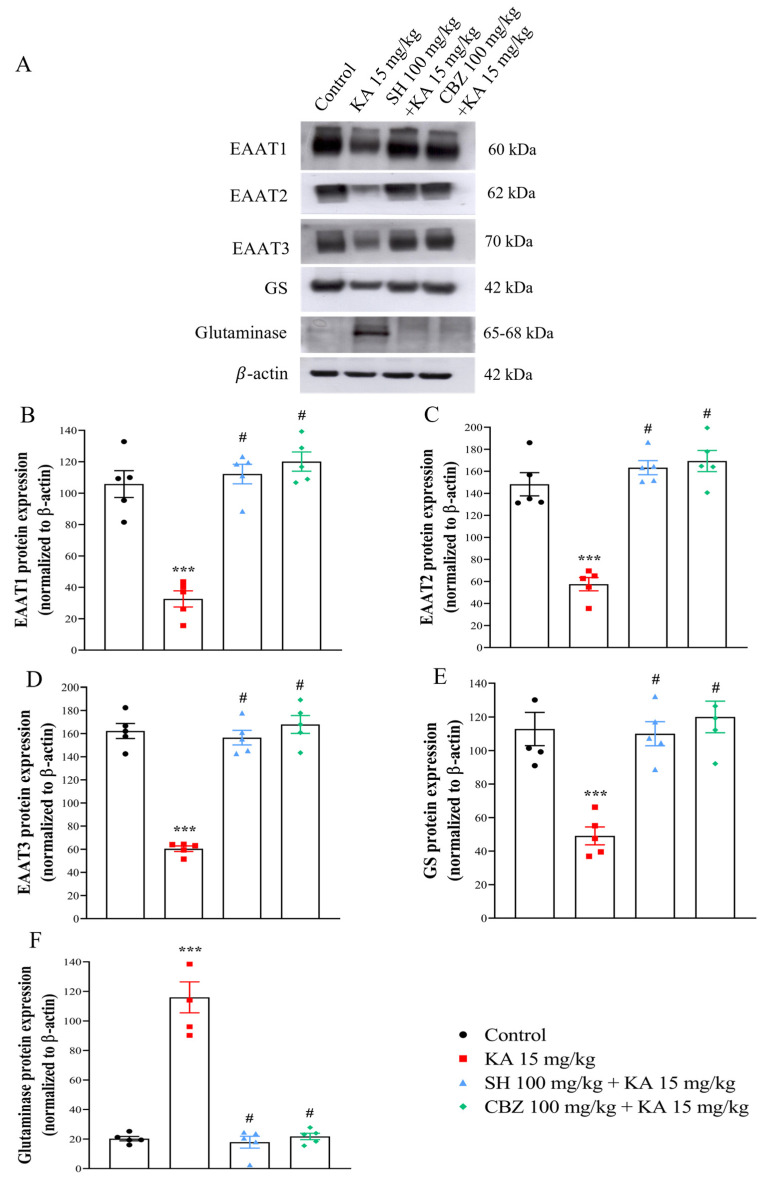
The protein expression levels of EAAT1, EAAT2, EAAT3, GS, and glutaminase in rat hippocampal tissue three days after KA injection. Representative Western blot images in the different groups (**A**) and densitometric values were normalized to β-actin (**B**–**F**). N = 5 animals per group. *** *p* < 0.0001 compared to the control group; # *p* < 0.0001 compared to the KA group.

**Figure 6 biomedicines-12-01312-f006:**
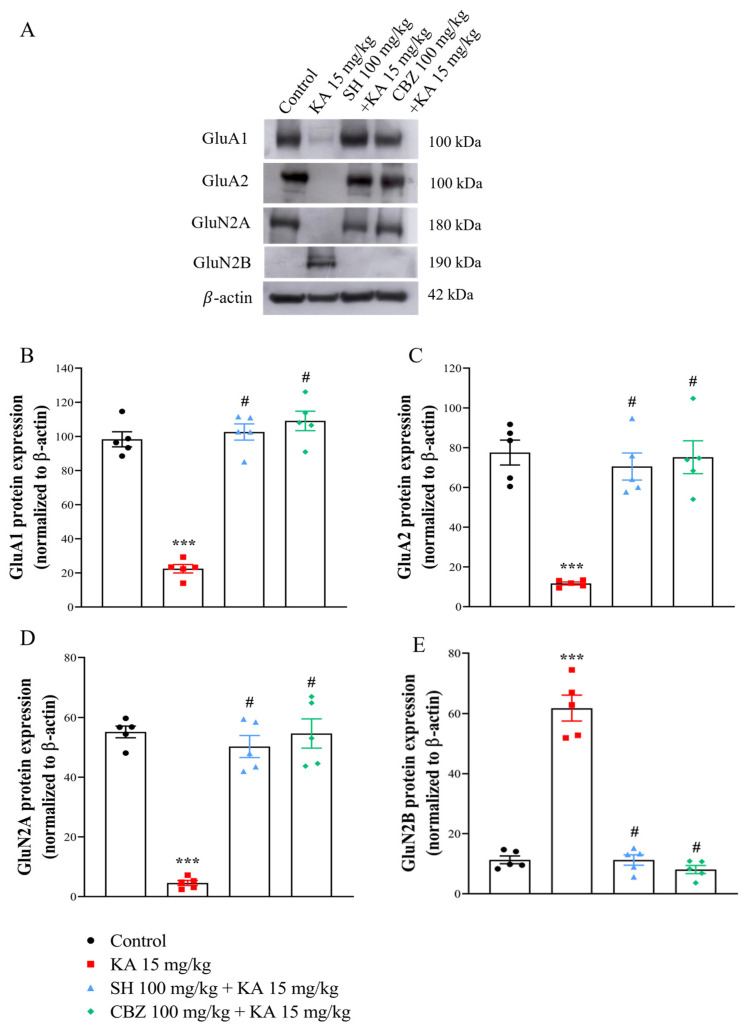
The protein expression levels of GluA1, GluA2, GluN2A, and GluN2B in rat hippocampal tissue three days after KA injection. Representative Western blot images in the different groups (**A**) and densitometric values were normalized to β-actin (**B**–**E**). N = 5 animals per group. *** *p* < 0.0001 compared to the control group; # *p* < 0.0001 compared to the KA group.

**Figure 7 biomedicines-12-01312-f007:**
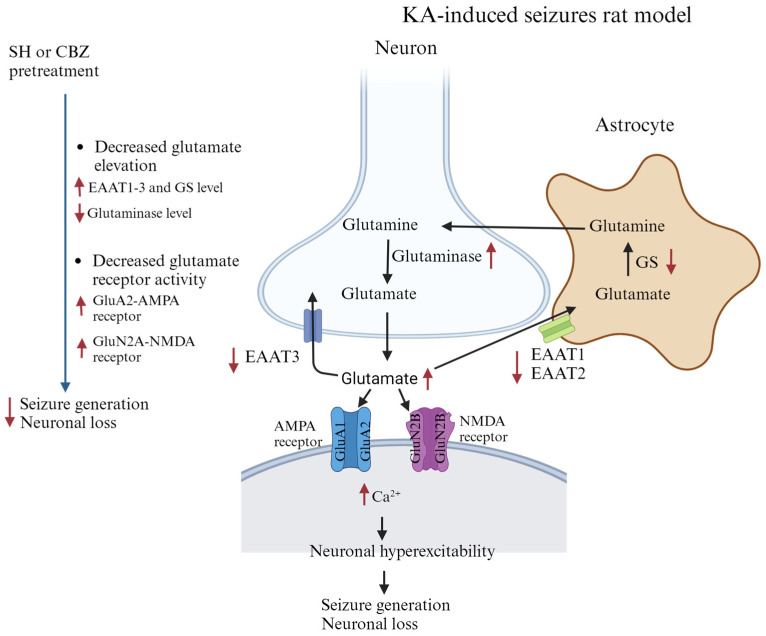
Schematic representation for the antiseizure and neuroprotective effects of SH in KA-induced rats. Graph created with BioRender.com.

**Table 1 biomedicines-12-01312-t001:** Effect of SH on seizures induced by KA in rats.

Group	Seizure Onset (min)	Seizure Score (Racine Scale)	% Seizure
KA 15 mg/kg	65.2 ± 3.9	4.5 ± 0.2	100% (11/11)
SH 50 mg/kg + KA 15 mg/kg	101.3 ± 15.8 **	3.9 ± 0.4	55% (5/9)
SH 100 mg/kg + KA 15 mg/kg	167	0.6 ± 0.5 ***	9% (1/11)
CBZ 100 mg/kg + KA 15 mg/kg	-	0.3 ± 0.2 ***	-

** *p* < 0.01, *** *p* < 0.0001 compared to the KA group.

## Data Availability

The raw data supporting the conclusions of this article will be made available by the authors on request.

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
