# Peer review of "Sodium Houttuyfonate Prevents Seizures and Neuronal Cell Loss by Maintaining Glutamatergic System Stability in Male Rats with Kainic Acid-Induced Seizures"

_biomedicines, 2024, doi:10.3390/biomedicines12061312_

Round 1

Reviewer 1 Report

Comments and Suggestions for Authors

This is an interesting carefully performed study about the anti-seizure effects of KH in a kainate model of epilepsy.

I have the following comments regarding only the interpretation of the results:

1. The authors claim neuroprotective effects of SH in this model (line 274). That is not necessary the case, since it might, similar like CBZ cause a lower insult caused by reduced seizure activity, which leads consequently to less neuronal damage. For evaluating neuroprotection the drug should have been given after the kainate injection.

2. Similarly, the interpretation of action of SH pretreatment needs to be reformulated (line 322 ff.). Since an experiment without kainate injection has not been presented it is difficult to state that the observed effects are a direct consequence of SH but not related to the reduced seizure activity.

3. Consequently, also the scheme (Fig. 7) needs to be changed. SH pretreatment should be replaced by seizure activity, the arrows should be adjusted and SH and CBZ would decrease the seizure activity.

Author Response

We thank the reviewer for the critical comments and constructive suggestions.

  1. The authors claim neuroprotective effects of SH in this model (line 274). That is not necessary the case, since it might, similar like CBZ cause a lower insult caused by reduced seizure activity, which leads consequently to less neuronal damage. For evaluating neuroprotection the drug should have been given after the kainate injection.

As suggestion by the reviewer, the sentence is modified toSH plays a preventive role in KA-induced seizure rats" (Page 11, line 287).

  1. Similarly, the interpretation of action of SH pretreatment needs to be reformulated (line 322 ff.). Since an experiment without kainate injection has not been presented it is difficult to state that the observed effects are a direct consequence of SH but not related to the reduced seizure activity.

According to this point, the group of rats treated only with SH is included in Fig. 2 and 3. The data are included in the result section (Page 5, lines 198-199).

  1. Consequently, also the scheme (Fig. 7) needs to be changed. SH pretreatment should be replaced by seizure activity, the arrows should be adjusted and SH and CBZ would decrease the seizure activity.

As suggestion by the reviewer, Figure 7 is modified (Page 12).

Reviewer 2 Report

Comments and Suggestions for Authors

Dear Dr. Yi Chang and co-authors, 

I've read your manuscript "Sodium houttuyfonate prevents seizures and neuronal cell loss by maintaining glutamatergic system stability in rats with kainic acid-induced seizures" with a great interest. I have several minor questions which could improve the quality of your manuscript: 

1. provide rationality for 3-days delay before collecting tissues for further histological, immunofluorescent and protein expression analyses in your experimental design  

2. you should have SH alone experimental group to probe its potential effects on neuronal proliferation 

3. Nissl/NeuN images analyses: a) how many regions of interest (ROI) per 1 slide have you assessed? b) did you manually label each neuron using Image J? c) can you analyze the intensity of NeuN+ signal instead? d) why did you assess only CA1 and CA3 sub-regions of the hippocampus? What about DG and CA2? The main reason is that SH+KA group (Figure 2) seems have more intense cresyl violet-stained signal, similar to NeuN-labelled (Figure 3) neurons. Additional hypothesis to your current one, is that SH may also stimulate neuronal proliferation, and eliciting thereby its protective effects on KA-induced seizures. 

4a. list of primary antibodies described in Methods, does not fit to the presented Results, e.g. PSD95, synaptophysin, GFAP, and CD11b are not presented on Figure 5. 

4b. what antibodies were used to probe GluA1, GluA2, GluNR1, GluN2A and GluN2B expression? Please, add it to the Methods. Again, GluNR1 is mentioned in the Results (p.9; line 235, 238), but no data for GluNR1 on Figure 6. 

 5. Discussion - please, discuss what are benefits of this new compound - SH (Sodium Houttuyfonate) in comparison with classical carbamazepine. 

Author Response

We thank the reviewer for the critical comments and constructive suggestions.

  1. provide rationality for 3-days delay before collecting tissues for further histological, immunofluorescent and protein expression analyses in your experimental design

As suggestion by the reviewer, the sentence ²The doses and schedules for drug administration were selected based on previous studies and pilot experiments [19, 21]" is added in the method section (Page 3, lines 96-97).

  1. you should have SH alone experimental group to probe its potential effects on neuronal proliferation

As suggestion by the reviewer, the group of rats treated only with SH is included in Fig. 2 and 3. The data are included in the result section (Page 5, lines 198-199).

  1. Nissl/NeuN images analyses: a) how many regions of interest (ROI) per 1 slide have you assessed? b) did you manually label each neuron using Image J? c) can you analyze the intensity of NeuN+ signal instead? d) why did you assess only CA1 and CA3 sub-regions of the hippocampus? What about DG and CA2? The main reason is that SH+KA group (Figure 2) seems have more intense cresyl violet-stained signal, similar to NeuN-labelled (Figure 3) neurons. Additional hypothesis to your current one, is that SH may also stimulate neuronal proliferation, and eliciting thereby its protective effects on KA-induced seizures.

We are interested in CA1 and CA3 of the hippocampus. We manually labeled each neuron using Image J. As suggestion by the reviewer, the number of surviving neurons and NeuN-positive cells in CA2 and DG is assessed. The data are included in the result section and supplemental Figure 1 (Page 5, line 192-197).

4a. list of primary antibodies described in Methods, does not fit to the presented Results, e.g. PSD95, synaptophysin, GFAP, and CD11b are not presented on Figure 5.

As suggestion by the reviewer, the sentence is modified to ² EAAT1 (1:10000; Abcam, Cambridge, UK), EAAT2 (1:50000; Abcam, Cambridge, UK), EAAT3 (1:10,000; Cell Signaling, Beverly, MA, USA), GS (1:50,000; Abcam, Cambridge, UK), glutaminase (1:10000), GluA1 (1:5000; Invitrogen, Waltham, MA, USA), GluA2 (1:5000; Invitrogen, Waltham, MA, USA), GluN2A (1:2000; Cell Signaling, Beverly, MA, USA), GluN2B (1:2000; Cell Signaling, Beverly, MA, USA), and b-actin (1:10000; Cell Signaling, Beverly, MA, USA)." (Page 4, lines 144-149).

4b. what antibodies were used to probe GluA1, GluA2, GluNR1, GluN2A and GluN2B expression? Please, add it to the Methods. Again, GluNR1 is mentioned in the Results (p.9; line 235, 238), but no data for GluNR1 on Figure 6.

As suggestion by the reviewer, the sentence is modified to ² EAAT1 (1:10000; Abcam, Cambridge, UK), EAAT2 (1:50000; Abcam, Cambridge, UK), EAAT3 (1:10,000; Cell Signaling, Beverly, MA, USA), GS (1:50,000; Abcam, Cambridge, UK), glutaminase (1:10000), GluA1 (1:5000; Invitrogen, Waltham, MA, USA), GluA2 (1:5000; Invitrogen, Waltham, MA, USA), GluN2A (1:2000; Cell Signaling, Beverly, MA, USA), GluN2B (1:2000; Cell Signaling, Beverly, MA, USA), and b-actin (1:10000; Cell Signaling, Beverly, MA, USA)." (Page 4, lines 144-149). In addition, the word ²GluNR1"is deleted from the result section (Page 10, line 253).

  1. Discussion - please, discuss what are benefits of this new compound - SH (Sodium Houttuyfonate) in comparison with classical carbamazepine.

As suggestion by the reviewer, the sentences ² In addition, although CBZ is a widely used antiepileptic drug, it has numerous side effects. SH is stable with satisfactory solubility and retains the major activities of H. cordata [46]. We infer that SH has the potential to be an effective and safe antiseizure drug." are added in the discussion (Page 13, lines 353-355).